# Observational Study of Microbial Colonization and Infection in Neurological Intensive Care Patients Based on Electronic Health Records

**DOI:** 10.3390/biomedicines13040858

**Published:** 2025-04-02

**Authors:** Alesya S. Gracheva, Artem N. Kuzovlev, Lyubov E. Salnikova

**Affiliations:** 1Federal Research and Clinical Center of Intensive Care Medicine and Rehabilitology, 107031 Moscow, Russia; palesa@yandex.ru (A.S.G.); artem_kuzovlev@fnkcrr.ru (A.N.K.); 2Vavilov Institute of General Genetics, Russian Academy of Sciences, 119991 Moscow, Russia; 3National Research Center of Pediatric Hematology, Oncology and Immunology, 117997 Moscow, Russia

**Keywords:** chronic patients with brain injury, intensive care unit, electronic health record, polymicrobial and monomicrobial colonization and infection, multidrug resistant organism

## Abstract

**Background/Objectives:** Patients with central nervous system injuries who are hospitalized in intensive care units (ICUs) are at high risk for nosocomial infections. Limited data are available on the incidence and patterns of microbial colonization and infection in this patient population. **Methods:** To fill this gap, we performed an electronic health record-based study of 1614 chronic patients with brain injury admitted to the ICU from 2017 to 2023. **Results:** Among the infectious complications, pneumonia was the most common (*n* = 879; 54.46%). Sepsis was diagnosed in 54 patients, of whom 46 (85%) were diagnosed with pneumonia. The only pathogen that showed an association with the development of pneumonia and sepsis in colonized patients was *Pseudomonas aeruginosa* (pneumonia: *p* = 7.2 × 10^−9^; sepsis: *p* = 1.7 × 10^−5^). Bacterial isolates from patients with and without pneumonia did not differ in pathogen titer or dynamics, but patients with monomicrobial culture were more likely to develop pneumonia than patients with polymicrobial culture (1 vs. 2 pathogens, *p* = 0.014; 1 + 2 pathogens vs. 3 + 4 pathogens, *p* = 2.8 × 10^−6^), although the pathogen titer was lower in monoculture than in polyculture. Bacterial isolates from all patients and all culture sites showed high levels of multidrug resistance (Gram-negative bacteria: 88–100%; Gram-positive bacteria: 48–97%), with no differences in multidrug-resistant organism (MDRO) colonization and infection rates. **Conclusions:** Our results highlight the high burden of MDROs in neurological ICUs and provide novel ecosystem-based insights into mono- and polymicrobial colonization and infection development. These findings may be useful for developing strategies to protect against infections.

## 1. Introduction

In recent years, there have been significant advances in the management of patients with severe central nervous system (CNS) injury [1,2]. However, most of these patients present with a variety of medical problems and long-term complications, are at high risk of developing critical illness, and require admission to the intensive care unit (ICU) [3,4]. The ICU is the hospital unit with the highest prevalence of healthcare-associated (hospital-acquired) infections (HAIs) [5]. ICU patients are susceptible to HAIs due to the high use of invasive procedures and devices, induced immunodepression, comorbidities, and the presence of multidrug-resistant organisms (MDROs) [6]. Patients with neurological disorders are at an even greater risk for HAIs than non-neurological patients because of their higher number of medical comorbidities, more frequent bedridden and immobilized status, higher prevalence and degree of immunodepression, and more frequent and prolonged use of external devices such as urinary catheters, central venous lines, mechanical ventilation, and prior use of antimicrobials [7,8]. The most common ICU-acquired infections are pneumonia (including ventilator-associated pneumonia (VAP)), surgical site or wound infections, bloodstream infections, and urinary tract infections, the latter two of which are often catheter-associated [9,10,11].

Airway protective reflexes are attenuated in patients with brain injury due to a decreased level of consciousness [12]. In addition to decreased mental status, patients experience multiple neural injuries, including dysphagia, impaired gag/cough reflexes, and a lack of secretion clearance [13]. These complications may increase the risk of ventilator-associated lung injury and nosocomial pneumonia [14] due to the need to protect the airway with endotracheal intubation and the initiation of mechanical ventilation. The incidence of ventilator-associated pneumonia (VAP) in patients with brain injury varies widely [15], reaching as high as 60% in some cases [16]. Brain injury itself may also predispose patients to bacterial pneumonia by mechanisms independent of mechanical ventilation or failure of airway protection [13,14]. CNS injury disrupts the normally balanced interplay between the immune system and the CNS, leading to secondary immunodepression (CNS injury-induced immunodepression, CIDS) and infection [17,18].

Patients with brain injury also have an increased risk of developing sepsis compared to the general ICU population. A multicenter Sepsis Occurrence in Acutely Ill Patients (SOAP) study reported that patients with brain injury were more likely to develop sepsis and respiratory failure than non-neurological patients, despite being younger and having fewer comorbidities [7]. Immune depression, malnutrition, systemic inflammation, dysregulated neurohormone secretion, and intestinal dysbiosis leading to dysregulation of the intestinal barrier and translocation of gut microbiota facilitate the development of sepsis in this patient population [19,20,21,22]. The incidence of sepsis in neurocritical care ranges from 1 to 62% [11], and 35–94% of sepsis cases develop in patients with nosocomial lower respiratory tract infections [23,24].

MDROs, defined as organisms with acquired non-susceptibility to at least one agent in three or more antimicrobial categories, have received considerable attention in pathogen research [25]. Antimicrobial resistance (AMR) is a major global health problem [26], and MDROs are responsible for high morbidity and mortality rates in ICUs [27,28]. However, the presence of potential pathogens and MDROs in microbiological specimens, even in debilitated and frail patients, is not proof of infection [6,29]. The increased use of microbiological specimens in the absence of well-defined clinical indications has been associated with the overuse of antibiotics. This in turn has led to a high prevalence of MDROs. Conversely, judicious use of antibiotics in conjunction with regular microbiological surveillance can provide pertinent data, thereby facilitating empirical antibiotic prescription [6,30].

It is imperative to have a comprehensive understanding of the unique characteristics of infection progression in microbially colonized patients. This knowledge is critical for preventing unwarranted antibiotic use and providing relevant, individualized data for judicious empiric antibiotic prescriptions. Frail and immunocompromised patients with brain injuries are particularly susceptible to infection and require individualized therapeutic regimens. However, there is a paucity of data in the literature regarding the nature and characteristics of microbial colonization and infection in this patient population. The main objectives of this electronic health record (EHR)-based study were to describe the characteristics of ICU patients with brain injury who did or did not develop pneumonia and sepsis and to compare their pathogen colonization and infection rates. This includes studying the predominant pathogen species in different cultures, the dynamics of pathogens and their titers, evaluating the role of polymicrobial colonization, and characterizing the prevalence of MDROs in the compared patient groups.

## 2. Materials and Methods

### 2.1. Study Design and Setting

Two types of analyses were performed: cohort and case-control nested within the full cohort. The cohort analysis presented data on the microbial spectra at different culture sites and their temporal patterns. The case-control analysis presented data on microbial colonization and the development of infections in neurological intensive care patients. The de-identified EHR data used in this study were obtained from the Russian Intensive Care Dataset (RICD). The RICD was developed by the Federal Research and Clinical Center of Intensive Care Medicine and Rehabilitology (FRCC ICMR) [31,32]. FRCC ICMR specializes in the treatment and rehabilitation of patients with severe brain damage due to vascular diseases (cerebral aneurysm, ischemic, and hemorrhagic stroke), brain injuries, brain tumors, and neurosurgical interventions for these and other diseases. In most cases, these patients have opportunistic and nosocomial infections due to their severe condition and prolonged stay in other hospitals. The EHRs contain data on all patients admitted to the ICUs of the FRCC ICMR between December 2017 and July 2023.

### 2.2. Participants

The study population included adults aged 18 years or older with central nervous system (CNS) injury who were admitted to and/or discharged from the intensive care unit (ICU) and had a minimum 24 h ICU stay. Participants were required to have at least one entry in the bacterial culture analysis module and the ICD-10 diagnosis module. We did not apply other inclusion/exclusion criteria, as they were already applied at the stage of the decision on the possibility of hospitalization at the FRCC ICMR, which was made by the Medical Commission of the FRCC ICMR in accordance with normative documents for compliance with the specialization of the FRCC ICMR. The following inclusion and exclusion criteria are employed for hospitalization in clinical yards, including the ICU. Patients who stand to benefit from admission are included; those with a poor prognosis and a high likelihood of death despite intensive care, or conversely, those deemed “too healthy” or requiring resources that cannot be provided, are excluded [33]. Some patients were hospitalized more than once at the FRCC ICMR; therefore, the patient identifier in this study was represented by the combined unique patient and hospitalization identifiers. Patients were stratified into groups with and without pneumonia and with and without sepsis, defined according to national and international consensus definitions for hospital-acquired pneumonia [34,35] and sepsis/septic shock [36].

### 2.3. Ethical Consideration

Data from the patients were collected from the EHR during the course of daily clinical practice. Informed consent was obtained from the patients or their relatives for the collection and processing of baseline data. Prior to the analysis, a de-identification process was implemented on all data. The establishment of this database, the anonymization technique, and the use of the data for research were approved by the local ethics committee of the Federal Research and Clinical Center of Intensive Care Medicine and Rehabilitology (No. 4/23/2, 20 December 2023). The creation of RICD did not affect clinical care and all medical information was protected. Therefore, the requirement for informed consent was waived for retrospective and observational RICD-based studies conducted by FRCC ICMR investigators, in line with other EHR-based studies in anesthesiology, resuscitation, and critical care [37,38,39,40].

### 2.4. Data Collection Procedures

The FRCC ICMR database, RICD, was developed based on advanced principles and methods used in international open database projects, such as the «eICU Program», «MIMIC-IV», and «MIMIC-III» [37,41,42]. The FRCC ICMR has initiated the implementation of a “digital” ICU, a concept that entails the aggregation and storage of all data from ICU patients on organizational servers, including continuously monitored parameters, thus ensuring its accessibility for analysis. RICD encompasses a wide range of patient data, including medical and anthropometric information, details of patient movement within the facility, diagnostic records, administered therapies, laboratory test results, scale assessments, vital and fluid parameters, and hospitalization outcomes. RICD comprises data on several vital parameters collected from bedside monitors and other equipment in the ICU, with up to 10 assessments per hour.

Data acquisition is described in detail in the first paper introducing the RICD database [31].

### 2.5. Microbiology

The European Committee on Antimicrobial Susceptibility Testing (EUCAST, https://eucast.org, accessed 5 January 2025) and Clinical and Laboratory Standards Institute (CLSI, https://clsi.org, accessed 5 January 2025) guidelines were followed for species identification and antimicrobial susceptibility testing. Microbial identification and antimicrobial susceptibility testing were performed using an automated system (BD Phoenix-100; Becton Dickinson and Company, Sparks, MD, USA), according to the manufacturer’s instructions. The list of antibiotics used in this study (*n* = 60), along with the concentrations tested for each antibiotic, is provided in Appendix A. The results were interpreted using the EUCAST clinical breakpoint tables for MIC based on their annual updates [43]. The antibiotic susceptibility classification was based on [44], where the “intermediate” category, which requires increased exposure to the drug, and the “sensitive” category, which requires normal exposure to the drug, were combined into one category opposite the “resistant” category. We used the MDR definition (non-susceptible to ≥1 agent in >3 antimicrobial categories) and lists of constructed antimicrobial categories for each organism or group of organisms from [25].

We did not establish lower bacterial load thresholds for specific culture sites because of the diversity of microbiological specimen types and sources and the nature of the study cohort, which was represented by frail and immunocompromised patients who are susceptible to infection. In addition, there was a high proportion of polymicrobial episodes, where even low levels of bacteria can affect outcomes [45].

### 2.6. Statistical Analysis

Calculations were performed using R software (version 3.4.1). For categorical variables, Fisher’s chi-square test was used to determine whether the proportion of the variable of interest was the same in the compared samples. Fisher’s chi-square test was preferred to logistic regression analysis for the following reasons. Logistic regression, like other asymptotic tests, assumes that the sample size is sufficiently large to provide an accurate estimate of the distribution of the test statistic under the null hypothesis. Multiple factors can interact to influence outcomes; however, including multiple factors in a logistic regression can lead to overfitting or overcorrection. The number of variables that can be included in the logistic regression can be estimated using the formula *n* = 100 + 50*i*, where *i* is the number of independent variables in the final model [46]. In our study, the sample size was relatively small when not evaluating the most abundant species, but the number of independent variables was very large and diverse. In addition, some important factors, such as the duration after brain injury in a wide range [47] and the use of hormonal and antimicrobial treatment before hospitalization, were not available and could not be included in the analysis, which reduced its strength.

For continuous variables, the normality of the distribution was assessed using the Shapiro–Wilk test. Because normality tests revealed violations of normality for some data samples, the Mann–Whitney U test (MWU test) was used to test whether two compared samples were pooled from the same sample. For multiple testing, FDR-corrected *p*-values < 0.05 were considered statistically significant.

Graphs were generated using https://www.bioinformatics.com.cn/srplot (accessed 10 December 2024), a free online data analysis and visualization platform. Logarithmic scales based on natural or decimal logarithms were used to improve the presentation and better assess the distribution ranges.

## 3. Results

### 3.1. Patient Characteristics

The patient characteristics are shown in Table 1. A total of 1614 ICU patients with bacteriological results were included in the study group. More than 90% of patients were transferred from other hospitals. Clinical scores, the Sequential Organ Failure Assessment (SOFA), and the Glasgow Coma Scale (GCS) were used to assess the severity of illness. An organ-specific SOFA score of three or higher is considered indicative of organ failure [48,49]. A GCS score of 3–8 is associated with severe brain injury, and 9–12 with moderate brain injury [50]. In our study group, the median SOFA score was 3, and 55% of the patients had moderate-to-severe brain injuries. The SOFA CNS subscale is based on the GCS score, so it is not surprising that the SOFA and GCS scores were inversely correlated (the scores have opposite directions): the Spearman rank coefficient was –0.67 (*p* < 1.0 × 10^−200^). Thus, the majority of patients in our study had moderate-to-severe nervous system complications of the primary disease, represented by cerebral infarction (>40%), followed by various neurological conditions, intracerebral hemorrhage, and traumatic brain injury. Among the diseases and conditions defined as complications of the primary disease, only mental disorder due to brain damage (in nearly 20% of patients) was clearly related to the primary disease; the remaining conditions were considered most likely to be consequences of the primary disease, and the three most common were flaccid neuropathic bladder, retinopathy and retinal vascular changes, and cerebral edema. The most common comorbidity was hypertensive heart disease with or without heart failure (59%). The mean number of microbiological tests per patient was 3.35, and more than 95% of patients had at least one positive bacteriological test. Pneumonia was diagnosed in >50% of patients; 15% of patients died, but sepsis and septic shock developed in less than 4% of patients.

### 3.2. Microbiology Results

Most bacteriological analyses were performed on respiratory biomaterials (*n* = 2808), of which 50% were endotracheal aspirates, 41% were bronchoalveolar lavage, 7% were sputum, and 2% were other samples. Urinary tract biomaterials (*n* = 1500) were routinely collected from urine or urine samples from urinary catheters. Blood cultures (*n* = 665) were obtained from venous blood from the peripheral vein (50%) or the catheter. Among the other small groups of culture sites (Table 1), wound samples (*n* = 155) were the most homogeneous, consisting of wound exudate (83%) or wound discharge. CSF analysis (*n* = 159) yielded only two positive cultures.

Figure 1 shows the top 10 pathogens, where available, for the main culture sites. Only the first isolate of a species from each patient was included in the analysis; duplicates (i.e., the same microorganisms found in consecutive samples) were excluded. The most commonly identified microorganism in the microbial communities was *Klebsiella pneumoniae* (range, 31–39%), depending on the site of culture. *Acinetobacter baumannii* was more common in respiratory samples (17%) and less common in the blood (9%) and urine (4%). *Proteus mirabilis* was common in all types of biomaterials. It is known to cause a variety of human infections (eyes, wounds, and gastrointestinal tract) but is most specific for catheter-associated urinary tract infections [51]. *P. mirabilis* is rarely considered a respiratory pathogen [52], so its high prevalence in respiratory samples (11%) was unexpected. In general, a relatively limited spectrum of the same microorganisms was found among the most common pathogens in all culture sites considered.

### 3.3. Temporal Pathogen Patterns in Respiratory and Urine Samples During the 30-Day Period

In ICUs, the turnover of hosts, transmission sites, and bacterial species over time leads to the establishment of specific bacterial communities, and knowledge of individual temporal patterns in multispecies communities may be useful in understanding the spread of infection. The patient-based comparison showed an overlap between respiratory and urinary pathogens. A total of 438 patients had the same species in both cultures, including 17% of respiratory samples and 32% of urine samples. The distribution of microorganisms during the 30-day period (3-day window) after admission is shown in Figure 2. Each pathogen had a unique temporal pattern. Early pathogens in respiratory samples are *S. aureus*, followed by *E. faecalis* and *A. baumannii*, while late pathogens are *P. stuartii*, *S. marcescens,* and *E. coli*. In urine samples, early pathogens are *A. baumannii*, followed by *E. faecalis* and *E. faecium*; while late pathogens are *P. stuartii*, *P. rustigianii,* and the fungus *C. albicans* (Figure 2, Appendix A). The differences shown in the heatmap matrices for the early and late pathogen patterns for each culture site are highly significant. For the seven pathogens common to both sites, the temporal dynamics in the respiratory and urine samples are similar (Appendix A). The time interval of colonization detection after admission did not correlate with the incidence; for example, the most common *K. pneumoniae* had a near-average temporal pattern.

### 3.4. Clinical and Microbiological Characteristics of Patients with and Without Pneumonia and Sepsis

The results of the case-control studies comparing pneumonia (*n* = 879) vs. non-pneumonia (*n* = 735) and sepsis (*n* = 54) vs. non-sepsis patients (*n* = 1560) by clinical characteristics are shown in Figure 3A and Appendix A. A large number of clinical characteristics associated with poorer health were also associated with the development of both pneumonia and sepsis. Patients with pneumonia were older and had worse SOFA and GCS scores than those without pneumonia. The majority of primary disease complications, several comorbidities, positive microbiological cultures, two or more colonization sites per person, sepsis, and death were more common in patients with pneumonia than in those without pneumonia. Sepsis was diagnosed in 54 patients, of whom 46 (85%) were diagnosed with pneumonia. The sepsis group was strongly enriched with patients with severe complications of the primary disease, which are not fully captured by severity scoring systems but worsen a patient’s condition, increasing the possibility of infection and weakening the ability to fight infection.

The distribution of pathogens in the groups showed a high prevalence of *P. aeruginosa* in respiratory samples from patients with pneumonia and sepsis compared to those without pneumonia or sepsis. *P. aeruginosa* was also strongly predominant in urine samples from patients with pneumonia, which also showed more prevalence of *E. faecium* and *C. albicans*. In patients with sepsis, the proportions of the species studied were similar to those in patients with pneumonia, but the results did not appear to be significant due to the sample power. The same explanation applies to less common species that were not detected in respiratory and urine samples from patients with sepsis. *S. aureus* and *E. coli* in respiratory samples and *P. mirabilis* and *E. coli* in urine samples were more frequently isolated from patients without pneumonia, possibly reflecting their lower antibiotic resistance (*S. aureus*) or being part of the normal commensal flora (*E. coli* and *P. mirabilis*).

### 3.5. Respiratory Pathogen Titers and Dynamics in Patients with and Without Pneumonia

The titer and dynamics of pathogens in the presence or absence of disease are illustrated using respiratory samples from pneumonia/non-pneumonia patients (Figure 4). The titer of pathogens in respiratory samples from patients with pneumonia was the same or even lower (*E. coli*, S. *maltophilia*, and *P. stuartii*) than that in patients without pneumonia (Figure 4A). The dynamics of the pathogens did not differ between the patient groups (Figure 4B). The violin plots for the temporal patterns generally show similar density peaks (a higher number of pathogen isolates) and troughs in each pathogen distribution over the study period in both patient groups. Some differences can be observed in the *P. aeruginosa* violin plot, with the opposite directionality of the peaks. Patients with pneumonia had a delayed peak compared to patients without pneumonia, who had an early peak followed by a gradual decline. This reflects the potential differences in *P. aeruginosa* colonization and infection rates between the groups. Figure 4C shows that the patterns of pathogen dynamics were largely similar in the compared patient groups; however, in the group without pneumonia, the relatively small contribution of *P. aeruginosa* to the microbial community tended to become infinitesimal over time.

### 3.6. Mono- and Polymicrobial Episodes in Patients with and Without Pneumonia

The microbiological pattern tended to be dynamic: monomicrobial episodes could alternate with polymicrobial episodes, and the composition and titer of pathogens could also vary between episodes. Therefore, the total number of samples collected during the entire hospitalization period was analyzed. We considered polymicrobial episodes as those in which pathogens were found in the same specimen regardless of the pathogen titer [45]. Pneumonia developed significantly more often as a result of monomicrobial than polymicrobial respiratory tract cultures in cases of infection with *K. pneumoniae*, *A. baumannii*, *P. mirabilis,* and *E. coli.* For the other pathogens, the direction of the effect was the same, except for *S. aureus* (Figure 5A). The incidence of pneumonia increased smoothly in a series of one to four pathogens. The significance of the differences in the odds of pneumonia when colonized with one versus two pathogens was 0.014 (OR = 1.24, 95% CI 1.05−1.46); with 1 + 2 pathogens versus 3 + 4 pathogens, 2.8 × 10^−6^ (OR = 2.25, 95% CI 1.60−3.16) (Figure 5B). The titer of pathogens in respiratory samples was significantly higher in polymicrobial compared to monomicrobial cultures (*K. pneumoniae, A. baumannii, P. mirabilis, S. marcescens, E. faecalis, P. aeruginosa*, and *P. stuartii*) (Figure 5C).

### 3.7. Multidrug-Resistant Bacterial Colonization in Patients with and Without Pneumonia and Sepsis

We compared the pathogen distribution with respect to MDR for the most common pathogens at the four culture sites (Figure 6). The antibiotics used to detect the MDR pathogens are listed in Appendix A. Data were available for 2406 bacterial isolates in the total group of 703 of 1614 patients included in the study. The fungi *C. albicans* and the bacteria *S. maltophilia* and *C. striatum* were excluded from the lists of common pathogens shown in Figure 1. *S. maltophilia* has intrinsic resistance to many categories of antibiotics but is treated with a high rate of success with trimethoprim-sulfamethoxazole (TMP-SMX). In our study, more than 75% of *S. maltophilia* isolates were susceptible to TMP-SMX, and there were insufficient data to define the MDR of *S. maltophilia*. There were also insufficient data for *C. striatum*.

A high prevalence of MDR bacterial isolates was found for all Gram-negative bacteria (GNB). This was observed in all cultures and in all patient groups. All wound cultures and all patients with sepsis, regardless of the culture site, were colonized with MDR GNB. The frequency of MDRO colonization did not differ between patients with and without pneumonia, with and without sepsis. The summary statistics for all patients and all cultures (Figure 6) for MDR GNB isolates are as follows: *P. stuartii* and *P. rustigianii,* 100% each; *K. pneumoniae,* 97%; *P. mirabilis* and *E. coli,* 96% each; *A. baumannii,* 94%; and *S. marcescens* and *P. aeruginosa,* 88% each. Thus, 88–100% of the GNB were represented by MDROs.

For Gram-positive bacteria (GPB), the frequency of MDR isolates also did not differ between pneumonia/non-pneumonia and sepsis/non-sepsis cases. The summary statistics for all patients and all cultures (Figure 6) for MDR GPB isolates are as follows: *E. faecium,* 97%; *S. aureus,* 62%; and *E. faecalis,* 48%. All *S. capitis* isolates (*n* = 12) were MDR, but these data may be biased because of the small number of observations.

### 3.8. Antimicrobial Resistance in Pathogens Prioritized by the World Health Organization

The World Health Organization (WHO) has prioritized MDR bacteria of public health importance that are resistant to last-resort antibiotics [26]. We supplemented the MDR data presented in Figure 6 with data available in our study for pathogens resistant to specific categories of antimicrobials belonging to the critical priority group, which included carbapenem-resistant Enterobacteriaceae, third- and fourth-generation cephalosporin-resistant Enterobacteriaceae, carbapenem-resistant *A. baumannii*, and a high-priority group, which included vancomycin-resistant *E. faecium* and carbapenem-resistant *P. aeruginosa* (Figure 7). In the previous phase of the study, we showed that antimicrobial MDR did not differ between patients with and without pneumonia and sepsis; therefore, the results are given for the entire patient sample. Not all, but some of the drugs in the defined categories (Appendix A) were tested, and an isolate classified as resistant had to be resistant to all drugs tested within an antimicrobial category.

High levels of AMR were identified for priority pathogens. Among the Enterobacteriaceae, the average prevalence of carbapenem-resistant *K. pneumoniae* from the considered culture sites was 75%, *P. mirabilis*, 36%; *E. coli*, 45%; *S. marcescens*, 51%; *P. stuartii*, 71%; and *P. rustigianii*, 100% (only two samples were available). Among *A. baumannii* and *P. aeruginosa* isolates, 89% and 85%, respectively, were resistant to carbapenems. The mean prevalences of third- and fourth-generation cephalosporin-resistant Enterobacteriaceae were as follows: *K. pneumoniae*, 83%; *P. mirabilis*, 55%; *E. coli*, 86%; *S. marcescens*, 60%; *P. stuartii*, 70%; and *P. rustigianii*, 100%. Of the *E. faecium* isolates, 72% were vancomycin-resistant.

## 4. Discussion

This study addressed the understudied problem of microbial colonization and infection patterns in chronic patients with brain injury admitted to the ICU. The results of this study revealed that the most common pathogens in the four culture sites (respiratory tract, urine, blood, and wound) were represented by a limited spectrum of the same microorganisms (Figure 1). These microorganisms exhibited specific temporal patterns (Figure 2). Pneumonia was observed in 54.46% of patients and was the most common infectious complication (Table 1). Sepsis was observed in 3.35% of the patients, with the majority of sepsis cases (85%) manifesting in those with pneumonia. Of the pathogens examined, only *P. aeruginosa* colonization was associated with the development of pneumonia and sepsis (Figure 3). Both GNB and GPB isolates from patients with and without pneumonia did not differ in pathogen titer or colonization dynamics, but patients with monomicrobial culture were more likely to develop pneumonia than those with polymicrobial culture. However, the titer was lower in the monoculture than that in the polyculture (Figure 4 and Figure 5). Bacterial isolates from all patients and all culture sites showed high levels of MDR; however, the incidence of MDROs did not differ between patients with and without pneumonia or sepsis (Figure 6).

The spectrum of bacteria most commonly encountered in our study is typical of ICUs [30,53]. Nosocomial GNB are the leading cause of ICU infections, with *K. pneumoniae* being the most common. Other common GNB include *E. coli*, *P. aeruginosa*, and *A. baumannii*. Among the GPB, *S. aureus* and Enterococcus species predominate [53]. Some differences from general ICUs still exist and are related to the fact that the patients in our study had severe neurological diseases that often required the use of urethral catheters. As a result, pathogens that more commonly colonize the urinary tract, such as *P. mirabilis* and *P. stuartii* (both in this and other studies [54,55]), were also among the predominant species in respiratory samples. In ICUs, microorganisms are organized in a specific persistent ecosystem that is highly resistant to extensive disinfection protocols [56,57]. This ecosystem is formed by the selection of a limited number of species that can outcompete native microbial communities by developing hypervirulence, resistance patterns, and the ability to form biofilms [27,58]. Bacterial communities have a hierarchical structure that includes temporal factors [59,60,61], which we can see in the example of our work as pathogen-specific unique temporal patterns observed in both respiratory and urine samples.

According to the literature, a specific bacterial etiology does not significantly influence the manifestations, course, and outcome of the disease, with age, comorbidities, and immune competence being more important [62,63]. Our data are generally consistent with these observations, except for *P. aeruginosa*, which was associated with a high incidence of infection in patients colonized by *P. aeruginosa*. In support of little, if any, pathogen-specific influence on the development of infection, our results compared the bacterial load and dynamics in the respiratory tract of ICU patients with and without pneumonia. None of the pathogens, including *P. aeruginosa*, had a higher titer in patients with pneumonia than in those without. Violin plots reflecting individual pathogen-specific dynamics and bar plots showing the contribution of each pathogen to the bacterial community during the 30-day period were largely similar in both patient groups, with major differences related to *P. aeruginosa* rates.

*P. aeruginosa* is the common etiologic agent of respiratory infections in the ICU [53]. This pathogen exhibits several transmission, survival, and intrinsic resistance mechanisms, including surface appendages such as type IV pili and flagella, which provide promising motility and adhesion capabilities; biofilm formation; quorum sensing (QS); viable but non-culturable (VBNC) state; and antibiotic resistance mechanisms, including efflux pump overexpression, reduced outer membrane permeability, and acquired or mutated resistance genes [64,65]. However, other common pathogens in this study also exhibit diverse adaptive mechanisms and antibiotic resistance determinants [26]; therefore, whether the overrepresentation of *P. aeruginosa* in patients with pneumonia is related to its higher affinity and pathogenicity than other pathogens in immunocompromised patients [66] or to the selection of extremely pathogenic clones, or was obtained by chance, is a challenge for future investigations.

The most interesting findings of our study are related to polymicrobial colonization of the respiratory tract and the development of pneumonia. Although many studies have considered polymicrobial events, they have mainly focused on the frequency of co-occurrence of specific pathogens and mechanisms of their interaction [45,67,68,69,70]. To the best of our knowledge, this study is the first to present a comprehensive comparison of polymicrobial and monomicrobial colonization events in terms of pathogen titer and infection development. Within polymicrobial infections, microbes exhibit competitive and cooperative mutual interactions mediated by metabolite exploitation, immune modulation, niche optimization, and virulence induction through competition for the same nutrient, cross-feeding of metabolites, production of other molecules that specifically harm or benefit other microbial community members, and signaling molecules [45,71]. Competition is thought to be the dominant type of interaction [72]. However, in the presence of antibiotics, clinical isolates often protect each other from clinically relevant antibiotics [71]. Bacterial resistance to the human immune system and to antibiotics is strongly associated with bacterial biofilm formation. Biofilms are responsible for 70% of all microorganism-induced infections and are major contributors to human HAIs [73]. All of the common pathogens considered in this study are biofilm producers. Compared to single-species biofilms, multi-species communities are characterized by changes in spatial organization and increased biomass, cell numbers, metabolic activity, and AMR [74]. An increased individual bacterial load in polymicrobial communities is a general phenomenon that has been demonstrated for all bacterial niches [45,67,75]. Although grouping is often beneficial, the physiological consequences of increased cell density in bacterial clusters can lead to increased competition. In biofilms, bacteria are enclosed in a self-produced biofilm matrix with limited outward diffusion and/or retention of compounds by the matrix. When cell densities are high and resources are low, bacteria compete for limited space and nutrients, for example, through QS-dependent regulation and other strategies regulated by positive feedback loops [76,77]. Our findings of higher individual bacterial titers and lower rates of progression from colonization to pneumonia for polymicrobial compared to monomicrobial communities are consistent with ecological models, suggesting that polymicrobial communities form an ecosystem that may be greater than the sum of its parts [71].

AMR is a major challenge in the ICU. In our cohort of neurological ICU patients, MDRO colonization/infection rates were higher than those described in the literature [29,53], despite cleaning procedures that included rigorous disinfection and sterilization of patient supplies and equipment during hospitalization. This may be due not only to the fact that the study cohort included chronically ill patients with a high percentage of external devices, but also to the high number of patients transferred from other clinics, who, along with staff, are known to be the most likely vectors of contamination in clinics [58]. There were no differences between MDRO colonization and infection rates, which is likely related to testing a broad spectrum of antibiotics and changing the antibiotic regimen in response to the test results.

Our study has several limitations. It was a retrospective, monocentric cohort, and therefore had inherent risks of limited accuracy and generalizability. The retrospective design inherently carries the risk of missing data and potential bias in data collection. Due to the retrospective nature of our study, we lacked data on the prior use of antibiotics and hormone therapy. Data on the presence of specific AMR mechanisms, such as methicillin-resistant *Staphylococcus* spp. and extended-spectrum β-lactamase/AmpC/carbapenemase-producing Enterobacterales, were also unavailable. Although the sample studied was not small, it was not large enough to account for a wide range of covariates when considering the pathogen strata. Despite these limitations, this study adds to the existing information and provides new insights into the peculiarities of bacterial colonization and infection in neurological intensive care patients.

## 5. Conclusions

Our study describes microbial colonization and infection (pneumonia and sepsis) in chronic patients with brain injury in ICU conditions and is the first to demonstrate that (i) each pathogen has a unique temporal pattern in a microbial ecosystem, including endo- and exogenous microorganisms from the patient and the ICU environment; and (ii) patients with monomicrobial cultures are more prone to develop infection than patients with polymicrobial cultures, although the pathogen titer is lower in monoculture than in polyculture. The only pathogen that appeared to be associated with the development of pneumonia and sepsis in colonized patients was *P. aeruginosa*. The pathogen titer, colonization dynamics, and the frequency of MDR isolates did not differ between patients with and without infection. Bacterial isolates showed higher MDR levels than those described in the literature.

The primary implications of this study are as follows. Except for *P. aeruginosa*, no specific microorganism was independently and significantly associated with an increased risk of developing infection after microbial colonization, suggesting that patient-specific factors are more important. The ICU environment fosters the development of a distinct ecosystem comprising a variety of microorganisms driven by exogenous and endogenous factors that regulate the colonization process according to their own dynamics and patterns. Infectious complications were associated with the presence of monoculture. Conversely, polyculture, even with high-titer pathogens, showed a protective effect due to competition for food and space resources and the production of inhibitory metabolites. This calls for renewed emphasis on the judicious and targeted use of antibiotics, with a focus on patient-centered approaches.

## Figures and Tables

**Figure 1 biomedicines-13-00858-f001:**
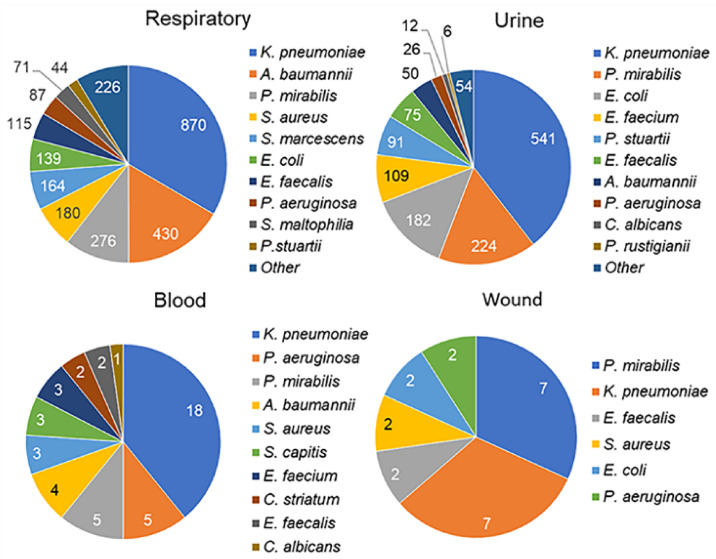
Species identified in microbial culture. Full names of microorganisms: *Acinetobacter baumannii*, *Candida albicans (fungus)*, *Corynebacterium striatum*, *Enterococcus faecalis*, *Enterococcus faecium*, *Escherichia coli*, *Klebsiella pneumoniae*, *Proteus mirabilis*, *Providencia rustigianii*, *Providencia stuartii*, *Pseudomonas aeruginosa*, *Serratia marcescens*, *Staphylococcus aureus*, *Staphylococcus capitis*, *Stenotrophomonas maltophilia*.

**Figure 2 biomedicines-13-00858-f002:**
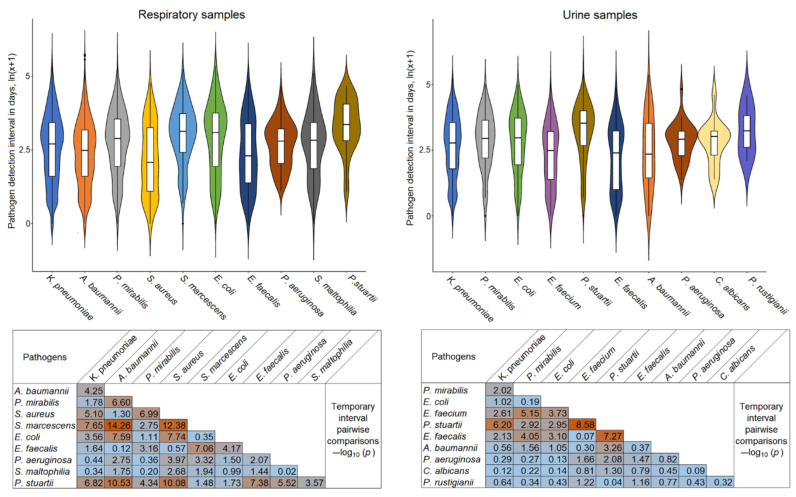
Temporal patterns of the top 10 respiratory and urinary pathogens over the 30-day period. The order of species is from largest to smallest, based on abundance. Violin plots for temporal distributions of microorganisms and heatmap matrices for *p*-values comparing these distributions are shown for respiratory (left) and urinary pathogens (right). The same species in the respiratory and urinary tracts are colored the same. Ten intervals within a 3-day window (0–29 days) are plotted on the Y-axis in ln(x + 1) format. All *p*-values < 0.05, and <0.022 (represented in the heat map matrix as (−log_10_) > 1.3 and > 1.66 for respiratory and urinary pathogens, respectively) are significant after FDR correction. The original data for the entire hospitalization period are presented in Appendix A.

**Figure 3 biomedicines-13-00858-f003:**
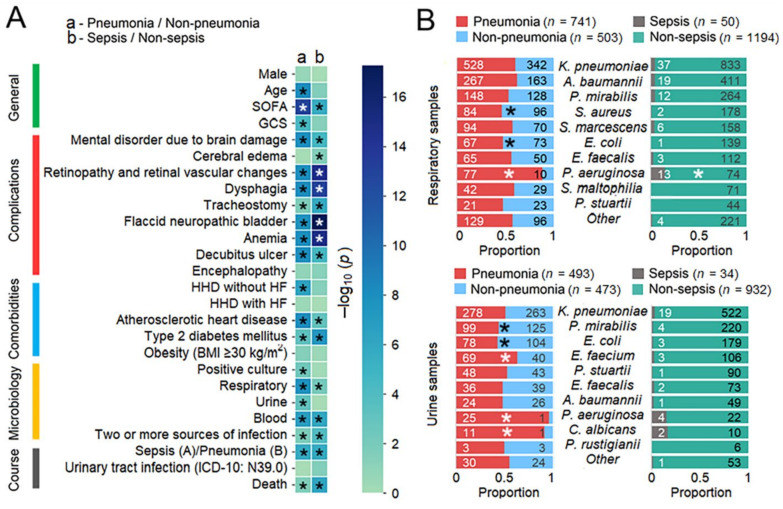
Results of clinical and microbiological comparisons of patients with and without pneumonia, with and without sepsis. (**A**) Heatmap of *p*-values for clinical variables in both the strata. (**B**) Distribution of the top 10 species between patients with and without pneumonia and with and without sepsis. In the legend, the number of patients with positive microbiological test results is given in parentheses. Significant *p*-values (**A**) and significantly different proportions (**B**) after FDR correction are marked with a *. Abbreviations: HHD, hypertensive heart disease; HF, heart failure.

**Figure 4 biomedicines-13-00858-f004:**
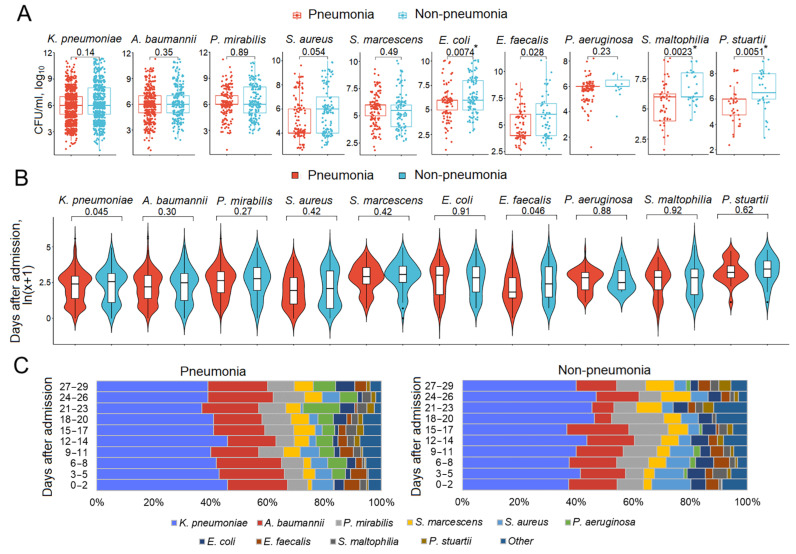
Patterns of respiratory pathogen titers and dynamics in patients with and without pneumonia. The order of species is from largest to smallest, based on abundance. (**A**) Pathogen titers during the entire hospital stay. Significant *p*-values after FDR correction are marked with a *. (**B**) Violin plots of the temporal distribution of microorganisms. Ten intervals within a 3-day window (0–29 days) are plotted on the Y-axis in ln(x + 1) format. (**C**) A 100% stacked bar shows the distribution of microorganisms relative to the length of hospital stay. Abbreviations: CFU, colony-forming units.

**Figure 5 biomedicines-13-00858-f005:**
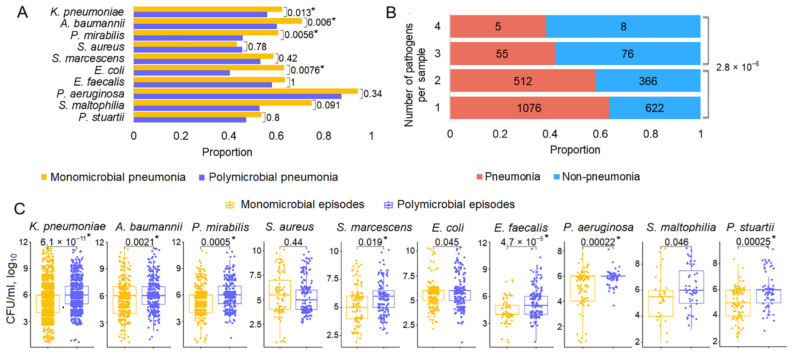
Monomicrobial and polymicrobial respiratory tract cultures. (**A**) The proportion of patients with pneumonia among the top 10 respiratory pathogens. (**B**) The proportion of patients with pneumonia in relation to the number of pathogens detected. (**C**) Pathogen titer in monomicrobial episodes compared with that in polymicrobial episodes. For polymicrobial cultures, combinations of the pathogen of interest with all the other pathogens were considered. (**A**,**C**) Significant *p*-values after FDR correction are marked with a *. Abbreviations: CFU, colony-forming units.

**Figure 6 biomedicines-13-00858-f006:**
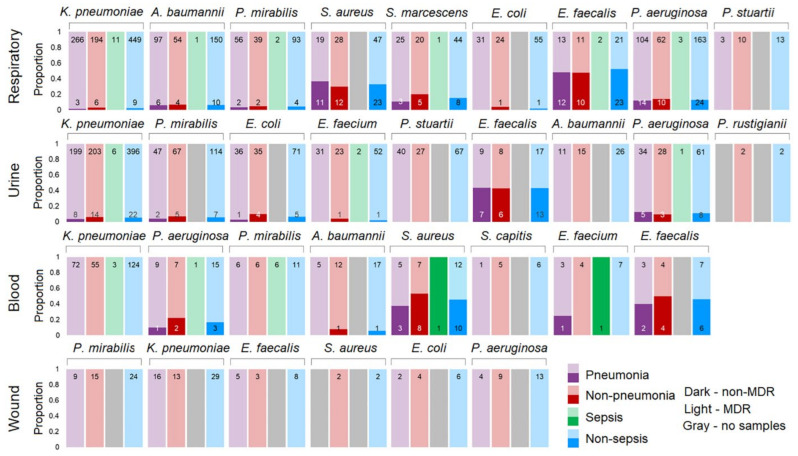
MDR and non-MDR bacterial isolates from patients in the ICU. For each pathogen of interest, data are presented for patient strata according to the order in the legend (lower right). Where samples were missing, to maintain the order, gray bars were included. Dark bars correspond to pathogens detected in non-MDR episodes, whereas lighter bars correspond to pathogens detected in MDR episodes.

**Figure 7 biomedicines-13-00858-f007:**
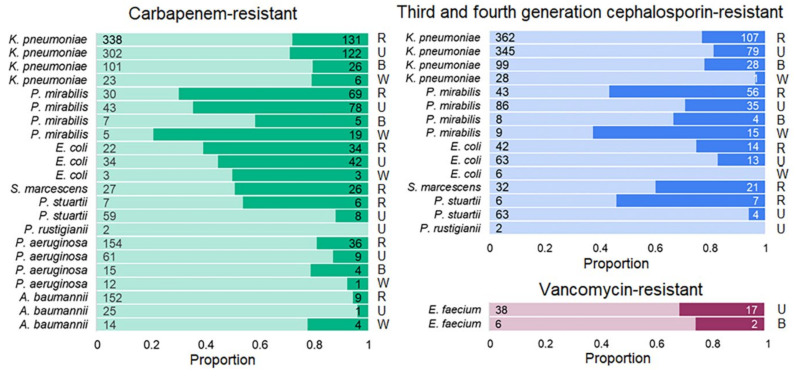
Critical and high-priority pathogens. Light bars correspond to pathogens that are resistant to the antibiotics of the last resort, while darker bars correspond to pathogens that are sensitive or of intermediate sensitivity to these categories of antibiotics. Abbreviations: R, respiratory; U, urine; B, blood; W, wound.

**Table 1 biomedicines-13-00858-t001:** Clinical presentation of ICU patients (*n* = 1614).

Characteristics	*n*	% or Me (IQR)
**General information**		
Male	919	56.94
Age	1614	61 (47–72)
Transfer patients	1487	92.13
Admission department—ICU	1577	97.71
Sequential Organ Failure Assessment, SOFA (0–24) *	1327	3 (2–5)
Glasgow coma scale, GCS (15–3) *	1355	11 (8–14)
**Primary disease**		
Intracerebral hemorrhage	294	18.22
Cerebral infarction	681	42.19
Intracranial injury	267	16.54
Other **	372	23.05
**Complications of primary disease**		
Mental disorder due to brain damage	316	19.58
Cerebral edema	286	17.72
Retinopathy and retinal vascular changes	320	19.83
Dysphagia	142	8.80
Tracheostomy	585	36.25
Flaccid neuropathic bladder	420	26.02
Anemia	188	11.65
Decubitus ulcer	197	12.21
Encephalopathy	118	7.31
**Concurrent diseases**		
Hypertensive heart disease without heart failure, HHD without HF	672	41.64
Hypertensive heart disease with heart failure, HHD with HF	285	17.66
Atherosclerotic heart disease	169	10.47
Type 2 diabetes mellitus	76	4.71
Obesity (≥30 kg/m^2^)	253	15.68
**Per-patient microbiology data**		
Positive culture	1545	95.72
Respiratory	1244	80.52
Urine	966	62.52
Blood	44	2.85
Eye, Ear, Nose, and Throat	16	1.04
Cerebrospinal fluid, CSF	2	0.13
Wound	13	0.84
Other	8	0.52
**Hospital course**		
Pneumonia	879	54.46
Urinary tract infection (ICD-10: N39.0)	36	2.23
Sepsis/Septic shock	54	3.35
Death	243	15.06
Discharge department: ICU	1035	64.13
Discharge department: Neurorehabilitation	285	17.66
Discharge department: Palliative psychiatric ward	294	18.22
ICU stay, days ***	998	25 (21–48)

Notes: Categorical data are expressed as n (%). Quantitative data are presented as medians (IQR). * The admission score range (in parentheses) is from best to worst. ** Subarachnoid hemorrhage; spinal cord injury; brain tumor, including after neurosurgical intervention for these and other conditions; *** For patients admitted to and discharged from the ICU.

## Data Availability

Availability of this data is limited. Data were obtained from RICD and are available at https://fnkcrr-database.ru (accessed on 6 February 2024) with the permission of Federal Research and Clinical Center of Intensive Care Medicine and Rehabilitology.

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
