# Peer review of "Observational Study of Microbial Colonization and Infection in Neurological Intensive Care Patients Based on Electronic Health Records"

_biomedicines, 2025, doi:10.3390/biomedicines13040858_

Round 1
Reviewer 1 Report
Comments and Suggestions for Authors
The authors performed a retrospective cohort study based on electronic health records of 1416 patients admitted to the neurological ICU from 2017 to 2023. The most common infectious disease was pneumonia (n=879), and 85% of sepsis cases (total n=54) occurred in patients with pneumonia. The only pathogen that showed an association with the development of pneumonia and sepsis in colonized patients was Pseudomonas aeruginosa. However, the results in the study seems very different.
Major concern
As per previous studies, A central nervous system (CNS) injury can significantly increase the risk of developing a bacterial infection due to the disruption of the blood-brain barrier, which normally protects the brain from invading pathogens, making it more susceptible to bacterial entry and subsequent infection; this can lead to serious complications like meningitis, brain abscesses etc.
https://emedicine.medscape.com/article/1806579-overview?form=fpf
https://www.nature.com/articles/nrn1765#:~:text=Infections%20are%20a%20leading%20cause,%2Dinduced%20immunodepression%20(CIDS).
The title may be revised according the results and data.
Abstract:
The whole abstract should be rewrite in a technical way while looking into some relevant published research articles in the journal. The methodology is very simples with no inclusion and exclusion criteria. The results have been presented in a non-technical and poor way.
Results in abstract should be revised for better understanding. Lines 14 and 15, [The most common infectious disease was pneumonia (n=879), and 85% of sepsis cases (total n=54) occurred in patients with pneumonia] should be revised as 85% may not be the representative of 879. This should also be revised in lines 230 and 231.
Line 16, [The only pathogen that showed an association with the development of pneumonia and sepsis in colonized patients was Pseudomonas aeruginosa] should be provided with statistically significant value along the statement in brackets.
Lines 22-23 could be benefited with some statistical values.
Keywords should be revised. Abbreviation may be removed from keywords.
Introduction
Line 71, MDROs should be clearly defined.
The last paragraph of introduction should contain a brief methods and results.
Methodology Section.
A clear inclusion and exclusion criteria should be provided at the start of methodology.
Line 84, if this was a case control study, then how many were cases and how many were control?
If there is any informed consent from the patient or not? It must be provided.
How many and which antibiotics susceptibility testing was performed? What was the concentration in testing? A brief procedure should be provided?
Results
Lines 314-324 should be revised and presented in a more understandable way.
Discussion
The first paragraph should be revised with some generalized problem statement and critical observations based on the title of study. Results may not be repeated in this section; however, figures and table may be used as a reference for comparison with previous study.
Line 373, should be revised and referenced.
The conclusion should be revised based on major findings.
Comments on the Quality of English LanguageNA
Author Response
Dear Reviewer,
Please see our answers to your questions in the attached file.
Best regards, Lyubov Salnikova

Reviewer 2 Report
Comments and Suggestions for Authors
The authors conducted a retrospective study based on electronic health records data to analyse the incidence of microbial colonization and infection in in neurological intensive care patients with central nervous system injury. This study provides relevant epidemiological results in the field of infection control accurately, by describing the patterns of microbial colonization and infectionin in such a vulnerable population. The study seem to be well conducted and the manuscript is well written. However, I have some suggestions to improve the contents’ organisation and then the redeability and quality of presentation. Please find the attachment.
biomedicines-3495884
Title
It should be informative about the study design.
Introduction
The introduction is clear and well-writting following a logical flow and the standards of scientific writing. I only suggest:
- Expanding and finishing the paragraph on MDROs (lines 71-76).
- Adding a new paragraph (that will be the last paragraph) on study rationale, including thesis statement, available evidence, knowledge gap and study novelty. These detail are important to introduce the study aim. Therefore, to this last paragraph you can join the study aims (lines 76-81).
Materials and Methods
Patients
The authors reported that they conducted a case-control study. Conversely, the authors stated that they conducted a retrospective study in the abstract section. Please clarify the study design. Further, I suggest adding a subsection labeled as “Study design and setting” where authors will describe the study design, the time frame, and the setting. After this section, the authors can report the “Participants” subsection where they describe the type of sampling and the elegibility criteria. Please be accurate in reporting inclusion and exclusion criteria.
I suggest adding a new subsection named as “Ethical consideration” where the authors will report any information regarding ethical aspects such as ethical approval number and date, consent form, and data confidentiality.
I suggest adding a subsection named as “Data collection procedures” where authors will describe how the data collection was undertaken and who collected the data.
Results
Table 1 should be improved. Please find below and example strcture (this is just an example of structure; please modify according to your data):
|
Table 1. Sample characteristics |
|||
|
N |
% |
||
|
Sex |
|||
|
Male |
79 |
45.4 |
|
|
Female |
95 |
54.6 |
|
|
Age |
|||
|
Years (mean; SD) |
61.9 |
13.52 |
|
|
Primary cancer |
|||
|
Breast |
19 |
10.9 |
|
|
Gastro-ent |
75 |
43.1 |
|
|
Brain |
2 |
1.1 |
|
|
Lung |
14 |
8.0 |
|
|
Genital |
7 |
4.0 |
|
|
Urinary |
4 |
2.3 |
|
|
Sarcoma |
3 |
1.7 |
|
|
Skin |
3 |
1.7 |
|
Discussion
Clear and well structured, addressing each study’s findings.
Conclusions
I suggest adding a paragraph about study’s implications.
Author Response

(The authors gave the same response as above.)

Round 2
Reviewer 1 Report
Comments and Suggestions for Authors
Minor technical corrections are still needed. See line 378 [A very strong predominance of MDR] The authors make sure the use of suitable word for every results presentation. Key words should be short and five as per journal criteria.
Comments on the Quality of English Languageminor correction are needed in grammar
Author Response
We thank the Reviewer for taking a close look at our work in order to make it as clear as possible for the reader.
We have edited the text very carefully regarding the use of the English language. Regarding specific comments, we have corrected the sentence beginning "A very strong predominance of MDR..." and reduced the number of keywords to five.
Lines 377-378: “A high prevalence of MDR bacterial isolates was found for all Gram-negative bacteria (GNB). This was observed in all cultures and in all patient groups.”
Keywords: chronic patients with brain injury; intensive care unit; electronic health record; polymicrobial and monomicrobial colonization and infection; multidrug resistant organism